# The Role of Exercise in Maintaining Mitochondrial Proteostasis in Parkinson’s Disease

**DOI:** 10.3390/ijms24097994

**Published:** 2023-04-28

**Authors:** Jingwen Li, Yanli Xu, Tingting Liu, Yuxiang Xu, Xiantao Zhao, Jianshe Wei

**Affiliations:** 1Department of Kinesiology, School of Physical Education, Henan University, Kaifeng 475000, China; 2Institute for Brain Sciences Research, School of Life Sciences, Henan University, Kaifeng 475004, China

**Keywords:** exercise, Parkinson’s disease, neurodegenerative disease, mitochondrial proteostasis, mitochondria

## Abstract

Parkinson’s disease (PD) is the second most common rapidly progressive neurodegenerative disease and has serious health and socio-economic consequences. Mitochondrial dysfunction is closely related to the onset and progression of PD, and the use of mitochondria as a target for PD therapy has been gaining traction in terms of both recognition and application. The disruption of mitochondrial proteostasis in the brain tissue of PD patients leads to mitochondrial dysfunction, which manifests as mitochondrial unfolded protein response, mitophagy, and mitochondrial oxidative phosphorylation. Physical exercise is important for the maintenance of human health, and has the great advantage of being a non-pharmacological therapy that is non-toxic, low-cost, and universally applicable. In this review, we investigate the relationships between exercise, mitochondrial proteostasis, and PD and explore the role and mechanisms of mitochondrial proteostasis in delaying PD through exercise.

## 1. Introduction

Parkinson’s disease (PD) is the second most common rapidly progressive neurodegenerative disease [1], affecting over 10 million people worldwide and leading to serious health and socio-economic consequences [2]. The main motor symptoms of PD are bradykinesia (slow movements), muscular rigidity, resting tremor, and abnormal posture and gait [3]. Its clinical manifestations include non-motor symptoms as well as motor symptoms, and the main pathology is characterized by the loss of dopaminergic neurons in the substantia nigra and the accumulation of α-synuclein (α-syn) misfolding [4,5].

Studies have shown that mitochondrial dysfunction is closely associated with the onset and progression of PD, and the use of mitochondria as a target for the treatment of PD has been gradually gaining in terms of both recognition and application [6,7,8]. In recent years, it has been observed that mitochondrial proteostasis is disrupted in PD patients, leading to mitochondrial dysfunction in the form of mitochondrial unfolded protein response (UPR^mt^), mitophagy, and mitochondrial oxidative stress (OS) [9,10,11]. Research in the fields of exercise and brain science have confirmed that appropriate physical activity is effective in delaying and improving PD [12,13,14,15]; however, there has been little research on whether mitochondrial proteostasis plays a key role in motor retardation and improving PD. Therefore, this study aims to determine out the relationships between exercise, mitochondrial proteostasis, and PD, as well as investigating the role of mitochondrial proteostasis in the delay PD through exercise and the related mechanisms.

## 2. Mitochondria and Mitochondrial Proteostasis

Mitochondria are double-membrane organelles found in almost all eukaryotic cells [16]. They consist of four parts—the outer mitochondrial membrane (OMM), the mitochondrial intermembrane space (IMS), the inner mitochondrial membrane (IMM), and the mitochondrial matrix—and are the main site of cellular adenosine triphosphate (ATP) production, which is involved in cell differentiation, growth, and signaling [17]. In addition to maintaining calcium homeostasis and synthesizing numerous cofactors, the mitochondria play important roles in the regulation of apoptosis, senescence, and innate immunity [18]. The functional diversity and adaptability of mitochondria depend on the synergistic action of numerous proteins and protein complexes [19]. Mitochondria are made up of over 1000 proteins, approximately 99% of which are encoded by the nucleus, translated by ribosomes in the cytoplasmic matrix, and ultimately imported into the mitochondrial matrix [20,21]. The remaining 13 proteins are encoded by mitochondrial DNA and are translated into the mitochondria [22]. The concept of mitochondrial proteostasis was introduced in the context of the mitochondrial matrix, where both cytoplasmic ribosomal-translated proteins and mitochondrial-translated proteins require proper folding and quality control [23]. Mitochondria are subjected to internal and external stimuli through UPR^mt^, mitophagy, and the regulation of related protein transport and mitochondrial OS to remove damaged and impaired mitochondria, regulate the body’s energy metabolism, and control the mitochondrial mass and quantity, thus maintaining the dynamics and homeostasis of mitochondria.

## 3. Mitochondrial Proteostasis in PD

The development of PD is often accompanied by an imbalance in mitochondrial proteostasis, which further exacerbates the course of PD. The maintenance of mitochondrial proteostasis consists of four main aspects: UPR^mt^, mitophagy, mitochondrial-associated protein transport, and mitochondrial OS. The specific regulatory mechanism is depicted in Figure 1.

### 3.1. UPR^mt^ in PD

UPR^mt^ is a mitochondrial stress response in which the mitochondria initiate a transcriptional activation program of nuclear DNA-encoded mitochondrial chaperone proteins and proteases to maintain mitochondrial proteostasis [8,24]. Mitochondrial chaperone proteins help to restore the normal conformation of misfolded proteins and the correct folding of newly synthesized proteins, mainly heat shock protein 60 (Hsp60), heat shock protein 9 (Hsp9), heat shock protein 10 (Hsp10), and heat shock protein 70 (Hsp70) [25]. Proteases that degrade irreversibly damaged or ineffective proteins include casein lytic proteinase P (CLpP), mitochondrial Lon protease 1 (LONP1), and YME1-like ATPase (YME1L1) [26,27]. When the mitochondrial matrix accumulates large amounts of unfolded, misfolded, and null proteins under various stress conditions, UPR^mt^ increases the expression of mitochondrial chaperone proteins and proteases, thereby promoting the restoration of mitochondrial proteostasis [28]. UPR^mt^ acts as a retrograde signaling pathway from the mitochondria to the nucleus, facilitating communication between the mitochondria and the nucleus [29].

UPR^mt^ regulates the onset and progression of PD. It has been found that disruption of activating transcription factor associated with stress-1 (ATFS-1), a major regulator of UPR^mt^, significantly reduced Parkinson’s disease-related-1 (PDR-1) mutants and accelerated the loss of dopamine neurons in PTEN-induced kinase 1 (PINK1) worms, while activation of UPR^mt^ in PDR-1 and PINK1 worms protected these mutants from the adverse effects of abnormal mitochondrial form and function, promoting the survival of dopamine neurons in PD [30]. In the PINK1 model of PD, ginsenoside proteins can exert neuroprotective effects by inducing the up-regulation of Hsp60, mitochondrial Hsp70 (mtHsp70), and CG5045 (a putative drosophila melanogaster protease that is 77% identical to worm CLpP)—which are key markers of UPR^mt^—in order to protect the mitochondria from PD-related OS [31]. However, it has been found that ATFS-1-mediated hyper-activation of UPR^mt^ induces non-apoptotic death of dopaminergic neurons [32]. Over-activation of the UPR^mt^ pathway enhances α-syn-induced dopaminergic neurotoxicity, whereas a deficiency of ATFS-1 attenuates α-syn proteotoxicity in dopaminergic neurons and animal muscle. These results suggest that ATFS-1-mediated overactivation of UPR^mt^ damages dopaminergic neurons and may exacerbate the α-syn pathogenesis in aged animals [33]. Thus, the activation of UPR^mt^ inhibits PD by promoting dopamine neuron survival, whereas excessive activation of UPR^mt^ is detrimental, as it damages dopaminergic neurons.

Based on the above, the rational activation of UPR^mt^ can promote the survival of dopamine neurons and exert neuroprotective effects to alleviate PD.

### 3.2. Mitophagy in PD

Mitophagy is a cellular degradation process that plays an important role in maintaining the mitochondrial population network and mitochondrial homeostasis [34]. Mitophagy is a form of autophagy that mediates the removal of defective or redundant mitochondria, and is the only known pathway that selectively removes entire mitochondria [35]. When mitochondria are stimulated, it causes an asymmetric division of the mitochondria, resulting in a weakening or loss of the membrane potential of the daughter mitochondria, producing depolarized mitochondria that enter the mitochondrial fusion re-circulation or are removed by means of the mitophagy pathway [36].

Altered mitophagy has been observed in almost all genetic or toxic environmental models of PD pathogenesis [37]. Degenerating dopaminergic mid-brain neurons in PD have been shown to experience elevated basal mitophagy in vivo, compared to other mid-brain neurons [38]. Excitatory calcium dysregulation in cortical neurons triggered by mutations in the dominant PD gene leucine-rich repeat kinase 2 (LRRK2) also led to increased mitophagy [39]. Additional evidence supporting the possible role of dysregulated mitophagy in neurodegeneration includes the observation that two cryptic PD genes and two LC3-interacting region (LIR) motif proteins associated with amyotrophic lateral sclerosis overlapping with frontotemporal dementia (ALS/FTD) are involved in the ubiquitin-mediated mitophagy pathway [40].

The PINK1/Parkin-mediated pathway is considered to be the most common and important pathway in mitophagy [41]. In the monogenic form of PD, the two genes that contribute to the autosomal recessive form of the disease are PARK2 (encoding the Parkin protein) and PARK6 (encoding the PINK1 protein), which encode Parkin and PINK1, respectively, both of which play important roles in PINK1/Parkin-mediated mitophagy. Dysfunctional mitophagy under reduced PINK1/Parkin binding mediates the pathological mechanisms of PD [42,43]. Up-regulation of PINK1/Parkin expression reduced reactive oxygen species (ROS) levels in SH-SY5Y cells and attenuated neurodegeneration in the brain tissue of PD mice, thereby promoting mitophagy and maintaining mitochondrial function in PD [44].

α-syn is a major component of Lewy bodies, and its mutation, duplication, or triplication leads to autosomal dominant PD. α-syn can impair mitophagy in several ways; for example, in the neurons of PD patients, α-syn accumulation leads to up-regulation of Miro protein levels on the outer mitochondrial membrane, which accumulates excessively and abnormally on the mitochondrial surface and delays mitophagy [45]. In A53T and E46K α-syn transgenic mice, α-syn accumulates on the mitochondrial membrane and promotes cardiolipin exposure on the mitochondrial surface. Cardiolipin exposure recruits the microtubule-associated protein 1 light chain 3 (LC3) to mitochondria and induces mitophagy [46]. These studies suggest that aberrant mitophagy plays a role in α-syn-mediated toxicity.

In addition to PARK6 and PARK2, other PD-associated genes, such as PARK7 (encoding the DJ-1 protein) and PARK8 (encoding the LRRK2 protein), have been reported to be mutated in patients with PD [47,48]. Both DJ-1 and LRRK2 proteins have been found to be associated with PINK1/Parkin-mediated mitophagy [49,50].

In summary, genes associated with PD play an important role in mitophagy, and mitophagy can influence the occurrence and development of PD, to a certain extent.

### 3.3. Mitochondrial Protein Import in PD

Mitochondria contain approximately 1000 different proteins, most of which are nuclear-encoded [51]. These nuclear-encoded mitochondrial proteins are synthesized as precursors in the cell membrane, and the mitochondrial import pathway brings the proteins into the OMM, IMS, IMM, and matrix [52]. Mitochondrial protein import is critical for mitochondrial health and function, and inadequate protein supply leads to imbalances in the mitochondrial internal proteome, the result of which is equivalent to defective mitochondrial genome expression [53]. On the other hand, import defects lead to the accumulation of precursor proteins in the cytosol and affect proteostasis outside the mitochondria [54]. Reduced import of mitochondrial proteins leads to the activation of UPR^mt^ [55]. However, limiting protein import also blocks the nuclear-encoded tricarboxylic acid (TCA) cycle and oxidative phosphorylation (OXPHOS) components, further reducing mitochondrial fitness [56].

The key protein complexes that regulate mitochondrial protein import are: (1) The translocase of outer membrane (TOM) complex; (2) the translocase of the inner membrane (TIM) channel complex; (3) the kinesin post-assembly modification (PAM) complex; (4) the mitochondrial processing peptidase (MPP) and matrix peptidasome Cym1/PreP; (5) the assembly machinery (SAM) outer membrane channel complex; (6) the melanoma inhibitory activity (MIA) inner and outer membrane space protein binding pathway; and (7) the missing in metastasis (MIM) outer membrane protein complex [57]. The normal function of these protein complexes contributes to the maintenance of mitochondrial proteostasis and to all vital activities of the body.

A study of post-mortem brain samples from α-syn transgenic mice found significantly reduced levels of TOM40, which has been reported to be a major target of PD [58]. Examination of post-mortem brain tissue from PD patients revealed abnormal α-syn-TOM20 interactions in nigrostriatal dopaminergic neurons, associated with a loss of import of mitochondrial proteins. In an in vivo model of PD, moderate knockdown of endogenous α-syn was sufficient to maintain the import of mitochondrial proteins. In addition, over-expression of the translocase of TOM20 or mitochondrial targeting signal peptide had a beneficial effect in the in vitro system, preserving mitochondrial protein import [59]. Inhibition of complex I (CI)—a characteristic pathological hallmark of PD—impairs the import of mitochondrial proteins, which has been associated with the down-regulation of two key components of this system: TOM20 and TIM23 [60]. Pathogenic mutations in Parkin have been shown to disrupt the interaction with TOM70 and TOM40. TOM7 is required to stabilize PINK1 at the outer membrane upon loss of membrane potential, and TOM70 has been shown to be required to import PINK1 into mitochondria [61,62]. Studies have also reported that α-syn over-expression is closely associated with reduced MIM expression [63].

Based on the above, PD is inextricably linked to mitochondrial protein import, which has emerged as a new strategy for the treatment of PD, as it occurs through mitochondrial membrane permeabilization production in the inner membrane and the process is an indicator of mitochondrial and cellular health status.

### 3.4. Mitochondrial Oxidative Stress in PD

Mitochondria are essential organelles in eukaryotic cells and are central to major cellular functions, including ATP production, cellular Ca^2+^ homeostasis, and regulation of ROS status [64]. ATP is produced in cells and involves the oxidation of substrates through cytoplasmic glycolysis, the tricarboxylic acid cycle, and OXPHOS [65]. OXPHOS is the main source of ATP in aerobic organisms, especially for neuronal cells with high energy requirements [66]. Furthermore, during OXPHOS, ROS are generated by the electron transport chain located in IMM. Moderate levels of ROS activate physiologically induced pathways (i.e., intracellular signaling cascades designed to maintain cellular homeostasis) that ensure normal function [67,68]. However, if there is an imbalance between the production/accumulation of ROS and the intrinsic antioxidant defenses that detoxify these reaction products, OS can occur, leading to irreversible cell and tissue damage and, ultimately, to disease [69].

Mitochondrial respiratory chain (MRC) impairment has been well documented in the substantia nigra of PD patients, as well as in various animal models. Considering the detrimental effects of ROS over-production and inadequate neuronal energy supply, maintaining MRC execution is critical to slowing the pathological process of PD [70,71,72]. The MRC is the basic structure of OXPHOS and plays a central role in cellular energy metabolism. The MRC consists of four enzyme complexes, including nicotinamide adenine dinucleotide ubiquinone reductase (NADH dehydrogenase; CI), ubiquinone succinate oxidoreductase (complex II; CII), ubiquinone cytochrome oxidoreductase (complex III; CIII) and cytochrome c oxidase (complex IV; CIV), as well as two mobile electron carriers, coenzyme Q (CoQ) and cytochrome c (Cytc) [73]. Increased stress due to ROS production is one of the proposed mechanisms of dopaminergic neuronal death in PD, and mitochondrial complex I is thought to be one of the main sources of ROS [36]. It has long been known that patients with idiopathic PD have reduced disease-specific mitochondrial CI activity or protein levels in the substantia nigra after death [74]. Many PD-related genes (e.g., PINK1, Parkin, DJ-1, LRRK2, MNRR1, SNCA, and VPS35) interact with or contribute to the assembly, phosphorylation, or normal activity of CI sub-units [75]. It has been found that CI activity in the substantia nigra and CI sub-unit levels in the striatum were reduced in PD patients. A decrease in CI has also been detected in the muscle and blood cells of PD patients. In addition, reductions in other MRC complexes have been reported in many PD patients [76]. 

Based on the above, it can be stated that PD is closely related to mitochondrial OS, and effective external interventions can regulate OS levels to protect the brain’s energy supply and delay the course of PD.

## 4. Exercise and Mitochondrial Proteostasis

Disruption of mitochondrial proteostasis leading to the progressive loss of cellular function is one of the hallmarks of aging [77]. Exercise is an effective means of slowing down aging and is effective in preventing or delaying many diseases caused by imbalances in mitochondrial proteostasis [78]. The specific regulatory mechanism is depicted in Figure 2.

### 4.1. Exercise and UPR^mt^

Aging reduces the expression of several mitochondrial genes in gastrocnemius muscle and is accompanied by low levels of UPR^mt^ markers, including YME1L1 and CLpP mRNA. In contrast, aerobic exercise increased the expression of UPR^mt^ marker proteins in skeletal muscle, including Hsp60, LONP1, and YME1L1 proteins in gastrocnemius muscle [79]. Exercise altered mitochondrial proteostasis and induced UPR^mt^ markers in the mouse hypothalamus, stimulating the production of OXPHOS components in mitochondrial DNA in neurons. The mechanism of stimulating mitochondrial genes in neurons through exercise to improve the control of energy homeostasis has emerged as an attractive target [80]. Moderate-intensity treadmill exercise induced UPR^mt^ in the white and brown adipose tissue of young rats [81]. Activating transcription factor 5 (ATF5) is a major regulator of UPR^mt^ in mammalian cells, which promotes the activation of downstream targets of UPR^mt^ during mitochondrial stress. Deletion of ATF5 resulted in increased mitochondrial ROS emission and mitochondrial chaperone protein expression, while acute exercise promoted ATF5 enrichment in the mitochondrial fraction. Furthermore, PPAR-gamma coactivator 1α (PGC-1α) was also identified as an additional regulator of basal expression of the UPR^mt^ gene [82,83]. These data suggest that UPR^mt^ may underlie the exercise-mediated effects of mitochondrial hormones.

### 4.2. Exercise and Mitophagy

Recycling of damaged mitochondria is tightly regulated by macroautophagy and mitophagy through the PINK1/Parkin signaling pathway, and can be stimulated by vigorous exercise [84,85]. Increased levels of the transcription factor EB (TFEB) protein—the major co-activator of autophagy-associated genes—and significant reductions in the autophagosome protein LC3 and the autophagic bridging sub-unit p62 protein after exercise marked a significant increase in autophagosome biogenesis and autophagy initiation [86]. Vigorous exercise also stimulates the recruitment of LC3-II, which induces mitophagy to remove dysfunctional organelle components from the muscle. Strenuous exercise also activates a key regulator of autophagy, autophagy-activated kinase 1 (Ulk1), and acute exercise-induced mitophagy may be regulated through the AMP-activated protein kinase (Ampk)–Ulk1 signaling axis [87]. Endurance training increases mitophagy in skeletal muscle, accelerates intracellular clearance of metabolic waste and damaged proteins, increases the conversion of LC3-I to LC3-II, increases the expression of the mitophagy receptors BCL-2 19 kDa interacting protein 3 (BNIP3) and Parkin, and improves mitochondrial quality by eliminating oxidative damage and damaged mitochondria through autophagy–lysosome interactions [88,89,90,91]. Exercise activates mitophagy, and appropriate exercise to maintain mitophagy at normal levels is essential for mitochondrial quality control.

### 4.3. Exercise and Mitochondrial Protein Import

Treadmill exercise increased TOM20 expression through the Caveolin-1 pathway and maintained input signaling, thereby protecting mitochondrial integrity and activating the expression of mitochondrial transcription factors, activating mitochondrial biosynthesis, and reducing brain ischemia-induced injury [92,93]. Exercise induces a large number of mitochondrial adaptations in the muscle, including mitochondrial protein import. Exercise has been shown to induce the expression of multiple mitochondrial protein import components [94], including the key protein import machinery components TOM20, TOM22, and TIM23, while increasing the rate of protein translocation to the mitochondria [95]. Exercise increases the activity of the protein import pathway, which drives the activation of UPR^mt^ and stimulates mitochondrial recycling through mitosis [96]. Thus, the up-regulation of components of the protein import machinery appears to be an important aspect of exercise-altered mitochondrial biogenesis.

### 4.4. Exercise and Mitochondrial Oxidative Stress

The levels of CIV and MRC complexes were higher in the skeletal muscle of older subjects who had undergone intense exercise training than those who had not, suggesting that exercise training improves mitochondrial function and the mitochondrial network structure in skeletal muscle of ageing humans [97,98]. Twelve weeks of resistance exercise training resulted in qualitative and quantitative changes in skeletal muscle mitochondrial respiration and moderate changes in mitochondrial proteins, particularly CI activity. The capacity for OXPHOS and electron transport systems is greatly reduced in sedentary animals, and the use of resistance exercise may prevent harmful conditions in mitochondrial skeletal muscle function [99]. Exercise-induced increases in CII activity may eliminate succinate, further reducing ROS production by platelet mitochondria, and ultimately inhibiting systemic OS in patients with cardiovascular disease. Furthermore, low-intensity blood flow-limiting resistance exercise increased coupled mitochondrial respiration in skeletal muscle to resist OS by increasing CII activity in patients with heart failure [100]. Exercise training significantly protected endothelial cells from oxidative damage caused by hyperhomocysteinemia and prevented the development of atherosclerosis by activating SIRT1 and inhibiting OS [101,102].

There is an upper limit to the amount of intensive exercise that can be performed without disrupting metabolic homeostasis, beyond which—for example, after a period of progressively harder training that exceeds expected physiological changes (i.e., over-training)—negative effects on metabolic health and physical adaptation begin to emerge, which appear to be caused by a partial shutdown of mitochondrial respiration and H_2_O_2_ production [103]. In animal models, moderate exercise increases the synthesis of OXPHOS complexes, improves mitochondrial energy efficiency, and improves brain mitochondrial bioenergetic function through mitochondrial enzyme induction. However, strenuous exercise leads to a significant reduction in OXPHOS CIV, a decrease in mitochondrial OXPHOS, and a decrease in ATP production [104]. Moderate exercise to maintain mitochondrial homeostasis at normal levels is, thus, essential.

## 5. Exercise Improves PD by Regulating Mitochondrial Proteostasis

The neuroprotective effects of physical exercise on PD have been extensively studied and are considered to be an effective intervention for improving PD [105,106]. It has been found that exercise can exert neuroprotective effects by modulating neurotrophic factors to support synaptogenesis and angiogenesis, alleviate OS, and improve mitochondrial function. It also delays PD by promoting cerebral blood flow and arousal, activating corticospinal excitability, and reducing intracortical inhibition [107]. A recent study has found that exercise may improve PD by regulating mitochondrial proteostasis [108]. The specific regulatory mechanism is depicted in Figure 3.

### 5.1. Exercise Improves PD by Regulating UPR^mt^

Regular exercise is considered to be an effective means of mitochondrial remodeling, stimulating mitochondrial biogenesis and promoting the dynamic balance of the body’s internal environment [109]. UPR^mt^ signaling pathways include the activating transcription factors 4 (ATF4)/activating transcription factors 5 (ATF5)–C/EBP homologous protein (CHOP) pathway, the Sirtuin-3 (Sirt3)–forkhead box O3a (FOXO3a)–superoxide dismutase 2 (SOD2) pathway, and the serine/threonine kinase (AKT)–estrogen receptor-alpha (Erα)–high-temperature requirement protein A2 (HtrA2) pathway [110]. Under basal steady-state conditions, CHOP can influence the mitochondrial composition by altering the correct stoichiometry of nucleus- and mitochondria-encoded proteins in the electron transport chain. Under basal conditions, a decrease in CHOP leads to a decrease in the expression of nuclear-encoded CIV [111]. In contrast, treadmill exercise improved the expression of CI and CIV in an MPTP-induced PD mouse model [63]. Not only is mtHsp70 the most critical protein of the Hsp70 family involved in mitochondrial protein transport, but it also interacts with the cofactor Hsp40 and the nucleotide exchange factor mitochondrial GrpE (MGE1) to bind unfolded substrate polypeptides and regulate mitochondrial protein folding [112]. The expression of mtHsp70 was significantly lower in MPTP mice than in controls, but increased significantly in the substantia nigra of mice after treadmill exercise, thereby improving mitochondrial proteostasis and delaying PD [63].

Therefore, it can be hypothesized that exercise regulates UPR^mt^ levels and maintains intracellular protein and mitochondrial proteostasis, thereby delaying PD.

### 5.2. Exercise Improves PD by Modulating Mitophagy

Studies have demonstrated the protective effect of exercise on mitochondrial dysfunction in PD [113], and have found that platform running improved symptoms in PD mice and increased mitophagy activity, as evidenced by reduced levels of the mitophagy detection proteins PINK1, Parkin, and p62 [114,115,116]. Aerobic exercise increased autophagy in the striatum of PD rats, promoted the expression of autophagy markers Beclin1 and LC3 II/I, and increased the expression of lysosomal degradation-related molecules such as lysosome-associated membrane protein 2 (LAMP2) and cathepsin L. Regular aerobic exercise affects and regulates the homeostasis of apoptosis and autophagy in the striatum and alleviates the neurodegenerative process of PD [116]. DJ-1 can compensate for PINK1 function, while DJ-1 mutations inhibit mitophagy, leading to the accumulation of obsolete mitochondria and inducing neurodegenerative lesions that, in turn, induce PD. The DJ-1 gene was up-regulated in the frontal cortex of rats after exercise, maintaining mitochondrial proteostasis and thereby delaying PD symptoms [117,118].

Based on the above, exercise can enhance mitophagy, improve autophagy disorders, and maintain mitochondrial proteostasis, thereby delaying PD.

### 5.3. Exercise Improves PD by Modulating Mitochondrial Protein Import

Regular exercise can limit the loss of dopaminergic neurons and improve mitochondrial dysfunction, thereby alleviating PD [63]. It has been found that the expression levels of TOM40, TOM20, and TIM23 were significantly reduced in the MPTP group of mice, and that increased expression of α-syn resulted in decreased expression of MIM-related proteins. Inactivation of MIM through α-syn accumulation blocks the import of mitochondrial precursor proteins responsible for mitochondrial structure and function [63]. The expressions of TOM40, TOM20, and TIM23 were significantly increased in mice after exercise, and the reduction of α-syn induced by treadmill exercise led to an increase in the expression of MIM-related proteins, thus increasing the import of mitochondrial precursor proteins and improving mitochondrial function [63]. Compared with the normal group, the levels of the mitochondrial fusion protein (OPA1, MFN2) and fission protein (Drp-1) in PD patients were reduced, while exercise improved the process of mitochondrial protein fusion and division in PD patients [119]. Thus, regular exercise improves mitochondrial membrane channel protein expression which, in turn, promotes mitochondrial protein transport function.

Based on the above, exercise can improve mitochondrial protein import function by improving mitochondrial channel proteins, as well as regulating protein fusion and fission balance, thereby delaying PD.

### 5.4. Exercise Improves PD by Modulating Mitochondrial OS

A study has found that treadmill exercise normalized the performance of CII–IV in the substantia nigra and striatum in a 6-OHDA-induced PD model, indicating that exercise maintains mitochondrial function and may reduce dopaminergic neuron damage [119]. Both CI and CIV were expressed at lower levels in the MPTP group, whereas both CI and CIV proteins were significantly elevated in PD mice after exercise, suggesting that the treadmill exercise-induced decrease in α-syn may have increased the import of proteins responsible for mitochondrial structure and function [63]. Treadmill training also improved CI expression and activity in PD neuromitochondria, suggesting a neuroprotective effect of treadmill exercise, which may be related to its benefits on mitochondrial biogenesis signaling and respiratory chain regulation in PD [120].

Therefore, exercise can alleviate OS damage and reduce mitochondrial dysfunction, thereby maintaining mitochondrial proteostasis and delaying PD.

## 6. Conclusions

Exercise can prevent and improve PD by regulating mitochondrial proteostasis. The specific mechanisms include the following: (1) Exercise maintains mitochondrial proteostasis by activating UPR^mt^; (2) exercise can improve mitophagy to clear damaged mitochondria and maintain mitochondrial proteostasis; (3) exercise regulates mitochondrial protein levels by regulating mitochondrial protein transport and maintaining mitochondrial proteostasis; and (4) exercise maintains mitochondrial proteostasis by alleviating OS damage and regulating the body’s energy metabolism. With continuous development in the field related to exercise and brain health, the prevention and treatment of various chronic diseases through exercise has gradually received widespread attention and recognition. With the progress of medical technology, human life expectancy continues to extend and the degree of social aging continues to deepen. The prevention and treatment of PD—which is highly prevalent in the elderly—has become a major challenge in the field of public health. Although there has been a great deal of research on the prevention and treatment of PD through exercise, there has been little systematic research on whether mitochondrial proteostasis plays a key role in delaying and improving PD through exercise, which is expected to facilitate the development of appropriate exercise prescriptions. Our future research should deepen the role and mechanism of mitochondrial proteostasis in delaying and improving PD through exercise in order to provide new ideas and a theoretical basis for the prevention and treatment of PD through exercise, as well as helping those in the medical field to develop relevant and rational exercise prescriptions.

## Figures and Tables

**Figure 1 ijms-24-07994-f001:**
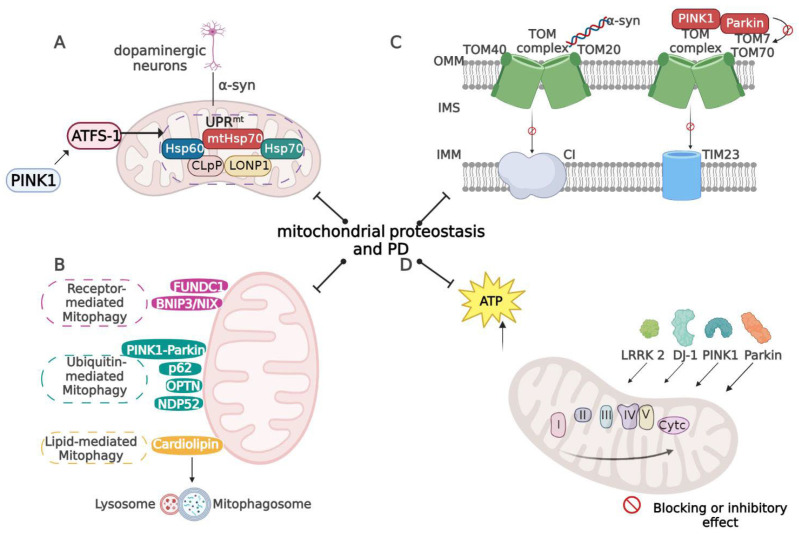
Mechanisms underlying the role of mitochondrial proteostasis in PD development: (**A**) Activation of UPR^mt^ attenuates PD and PINK1 activates ATFS-1-dependent UPR^mt^, which then promotes dopaminergic neuron survival; (**B**) major mechanisms of Parkin-dependent and Parkin-independent mitophagy; (**C**) in PD, α-syn binds Tom20 and reduces precursor import, which is manifested as CI dysfunction and elevated ROS levels. The deletion of TOM40 in PD patients and models is associated with the accumulation of α-syn. In addition, PINK1/Parkin-mediated mitophagy requires TOM7 and TOM70; and (**D**) PD-related genes (e.g., PINK1, Parkin, DJ-1, LRRK2) interact with the OXPHOS complex (created using Biorender.com).

**Figure 2 ijms-24-07994-f002:**
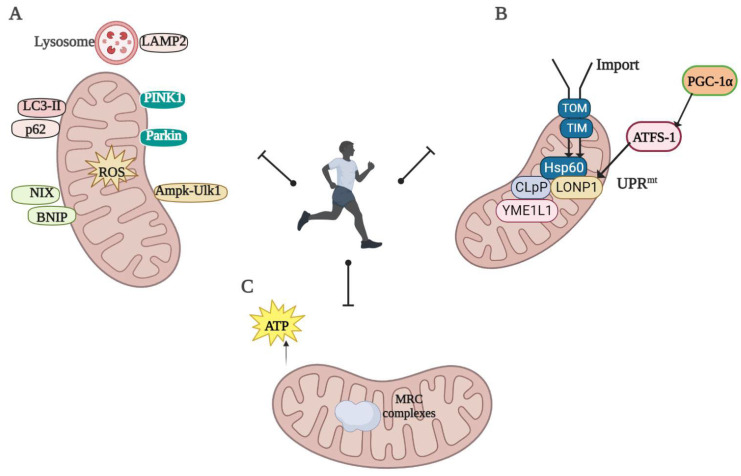
Mechanisms of regulation of mitochondrial proteostasis by exercise: (**A**) Increased levels of the autophagosomal proteins LC3 and p62 protein after exercise activate Ulk1, a key regulator of autophagy, which is regulated by the Ampk–Ulk1 signaling axis. Increased mitophagy, conversion of LC3-I to LC3-II, increased expression of mitophagy receptors BNIP3 and Parkin, and improved mitochondrial quality by eliminating oxidative damage and damaged mitochondria through autophagy–lysosome interactions; (**B**) exercise increases the expression of UPR^mt^ marker proteins, including Hsp60, LONP1, and YME1L1 proteins. ATF5 promotes the activation of UPR^mt^ downstream targets during mitochondrial stress. PGC-1α has also been identified as an additional regulator of basal UPR^mt^ gene expression. Exercise increases the activity of the protein import pathways TOM20, TOM22, and TIM23, which drive the activation of UPR^mt^ and stimulate mitochondrial recycling through mitosis; and (**C**) exercise can alleviate OS damage and reduce mitochondrial dysfunction (created using Biorender.com).

**Figure 3 ijms-24-07994-f003:**
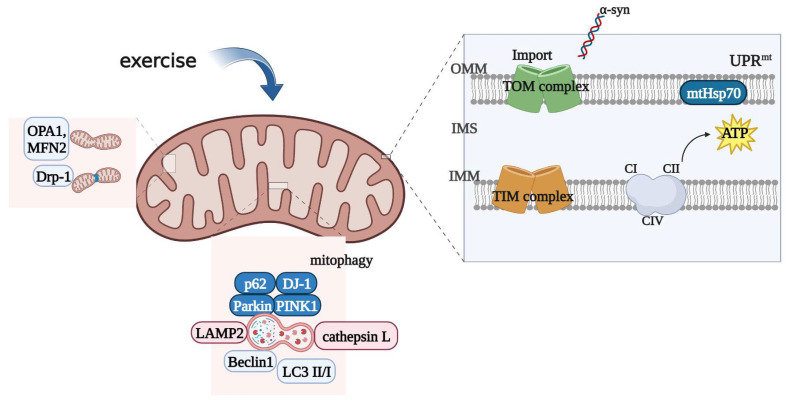
Mechanisms by which exercise improves PD through mitochondrial proteostasis. In a mouse model of PD, exercise increased the expression of CI, CIV, and mtHsp70 and improved mitochondrial protein stability. Exercise decreased the expression of the mitotic detection proteins PINK1, Parkin, and p62, as well as the autophagy markers Beclin1 and LC3 II/I, and increased the expression of lysosomal degradation-related molecules such as LAMP2 and cathepsin L. Up-regulation of the DJ-1 gene maintained mitochondrial protein stability. Exercise increased the expression of TOM40, TOM20, and TIM23, while the induced decrease in α-syn led to the increased expression of MIM-related proteins, increased input of mitochondrial precursor proteins, and improved mitochondrial function. Exercise improved the process of mitochondrial protein fusion and division in PD patients, as well as the expression of mitochondrial membrane channel proteins, which, in turn, promoted the transport function of mitochondrial proteins. Exercise normalized the expression of CI–IV and increased cytochrome c concentration and ATP production in a PD model (created using Biorender.com).

## Data Availability

Not applicable.

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
