# Peer review of "The Role of Exercise in Maintaining Mitochondrial Proteostasis in Parkinson’s Disease"

_ijms, 2023, doi:10.3390/ijms24097994_

Round 1
Reviewer 1 Report
The study present the relationship between exercise, mitochondrial proteostasis, and PD, and to investigate the role of mitochondrial proteostasis in exercise delay PD and its mechanisms. The work is practical and interesting.
Author Response
We would like to thank the reviewers for their positive comments on the manuscript. We have revised and edited the manuscript in English.
Reviewer 2 Report
Dear Doctor Li!
I have an honor to read and review your manuscript “The Role of Exercise-induced Mitochondrial Proteostasis in Parkinson's Disease”
The work is devoted to the most important problem of studying the mechanisms underlying Parkinson's disease and finding ways to correct this disease. The paper presents an in-depth description of the mechanisms of mitochondrial proteostasis and their disorders in Parkinson's disease, the effect of exercise on their correction.
The work is written in a good scientific language, well structured. I ask you to pay attention to the rules for using abbreviations (Line 88 where the abbreviations of pdr-1 and PINK, LIR motif proteins aе Line 118, ALS-FTD at Line 119 first appear without decryption), and I kindly ask you to correct it and carefully check the text for the presence of abbreviations without decryption.
When reviewing the work, I had the following fundamental question.
You rightly mention the role of oxidative stress in the pathogenesis of Parkinson's disease (Line 92-93). But in the main theses of your manuscript, the process of mitochondrial oxidative phosphorylation is mentioned as one of the consequences of disruption of mitochondrial proteostasis. Here, in my opinion, there is a fundamental confusion of concepts. Oxidative phosphorylation is one of the main NORMAL functions of the mitochondria and cannot in itself be a manifestation of mitochondrial dysfunction, since it is a normal physiological process. Another case is that when ensuring the normal coupling of oxidation and phosphorylation in the respiratory chain of mitochondria, an inevitable leakage of electrons occurs (within 3% of the total amount), which leads to the synthesis of reactive oxygen species (ROS), primarily superoxide and peroxide radical. ROS play an important role in the processes of intracellular signaling (RedOx regulation), and a violation of their synthesis (an increase of ROS production), together with the depletion of antioxidant defense systems, leads to oxidative stress (OS) and its consequences in the form of a violation of RedOX regulation and damage of biopolymers (including DNA, primarily mitochondrial DNA as it is not protected chaperone proteins, and proteins). In your work, sufficient attention is paid to the connection of disorders in the respiratory chain complexes (especially COX 1) with Parkinson's disease. But these disorders are the basis of OS formation. Please explain what you meant when you say that OXPHOS is a consequence of violations of mitochondrial proteostasis - perhaps it is necessary to replace OXPHOS with OS in the terminology and the corresponding text? (Lines 15, 37, 68). Or you’ve mean OXPHOS violations, depletions, decrease?
At Line 188-193 you are writing about electron transfer through Respiratory chain (OXPHOS pathway) which results in ROS synthesis. I am afraid it louds not very clear, because the main function of electron transport is not the ROS synthesis, but more complex processes leading to the changing in Gibbs free energy between COX, which makes ATP synthesis possible, when oxygen is the acceptor of electrons, which are needed for this reactions, and electrons leakage from the respiratory chain is the source of ROS as a by-product of OXPHOS. At the same time, the loss of electrons and the formation of ROS occurs under normal conditions, without damage to mitochondria and the formation of OS. In my opinion, this process can be described in your work more accurately.
You are writing about (Line 211-212) that effective external interventions can improve OXPHOS levels, protect the brain's energy supply and slow down the course of PD.
It is known that activation of tissue respiration in the respiratory chain leads to an increase not only in ATP synthesis, but also in ROS production. An increased level of ROS can, on the one hand, strengthen certain mechanisms of RedOx regulation, but, on the other hand, cause the development of OS. Please explain what level of improvement of OXPHOS you mean. An increase in weakened OXPHOS with PD to a normal level? At Line 310 you claim that exercise can inhibit OS. But what is the underlying mechanism of OS inhibition if OXPHOS is activated after exercises and ROS production must be increased, too?
Exercises restores OXPHOS capacity and activates COX protein activity (Line 303-304, 324). Where is the measure when exercises are useful, and where do they become harmful (how is overtraining, excessive exercise, fatigue related to the state of mitochondria and proteostasis?) I think that a discussion of these issues would be useful in your article.
Author Response
Response to Reviewer’s Comment
First of all, we would like to thank the reviewers for their invaluable suggestions and positive comments on the manuscript. We carefully examined these comments and have revised the manuscript point-by-point along the lines suggested.
The corrections in the revised manuscript are marked using the 'track changes' function.
Point 1: I ask you to pay attention to the rules for using abbreviations (Line 88 where the abbreviations of pdr-1 and PINK, LIR motif proteins aе Line 118, ALS-FTD at Line 119 first appear without decryption), and I kindly ask you to correct it and carefully check the text for the presence of abbreviations without decryption.
Response 1: We have rechecked the abbreviations in the manuscript and amended them in accordance with the rules for abbreviations. For example, Line 48 adenosine triphosphate (ATP), Line 88 Parkinson's disease-related-1 (PDR-1), Line 89 PTEN-induced kinase 1 (PINK1), Line 93 mitochondrial Hsp70 (mtHsp70), Line 120 leucine-rich repeat kinase 2 (LRRK2), Line 123 LC3-interacting region (LIR), Line 123 amyotrophic lateral sclerosis overlapping with frontotemporal dementia (ALS/FTD), Line 143 microtubule-associated protein 1 light chain 3 (LC3), Line 167 post-assembly modification (PAM), Line 169 assembly machinery (SAM), Line 170 melanoma inhibitory activity (MIA), Line 171 missing in metastasis (MIM), Line 182 complex I (CI).
Point 2: You rightly mention the role of oxidative stress in the pathogenesis of Parkinson's disease (Line 92-93). But in the main theses of your manuscript, the process of mitochondrial oxidative phosphorylation is mentioned as one of the consequences of disruption of mitochondrial proteostasis. Here, in my opinion, there is a fundamental confusion of concepts. Oxidative phosphorylation is one of the main NORMAL functions of the mitochondria and cannot in itself be a manifestation of mitochondrial dysfunction, since it is a normal physiological process. Another case is that when ensuring the normal coupling of oxidation and phosphorylation in the respiratory chain of mitochondria, an inevitable leakage of electrons occurs (within 3% of the total amount), which leads to the synthesis of reactive oxygen species (ROS), primarily superoxide and peroxide radical. ROS play an important role in the processes of intracellular signaling (RedOx regulation), and a violation of their synthesis (an increase of ROS production), together with the depletion of antioxidant defense systems, leads to oxidative stress (OS) and its consequences in the form of a violation of RedOX regulation and damage of biopolymers (including DNA, primarily mitochondrial DNA as it is not protected chaperone proteins, and proteins).
Response 2: We wanted to highlight in this section that ROS imbalance can lead to OS, leading to irreversible cell and tissue damage and ultimately disease, and therefore made the following refinements and modifications.
" OXPHOS is the main source of ATP in aerobic organisms, especially for neuronal cells with high energy requirements [66]. Furthermore, during OXPHOS, ROS are generated by the electron transport chain located in IMM. Moderate levels of ROS activate physiologically induced pathways (i.e., intracellular signaling cascades designed to maintain cellular homeostasis) that ensure normal function [67, 68]. However, if there is an imbalance between the production/accumulation of ROS and the intrinsic antioxidant defenses that detoxify these reaction products, OS can occur, leading to irreversible cell and tissue damage and, ultimately, to disease [69]." (Line 198-206).
Reference:
66.Cheng, X. T.; Huang, N.; Sheng, Z. H., Programming axonal mitochondrial maintenance and bioenergetics in neurodegeneration and regeneration. Neuron 2022, 110, (12), 1899-1923.
67.Shen, Y. H.; Wu, Q.; Shi, J. S.; Zhou, S. Y., Regulation of SIRT3 on mitochondrial functions and oxidative stress in Parkinson's disease. Biomedicine & Pharmacotherapy 2020, 132.
68.Imbriani, P.; Martella, G.; Bonsi, P.; Pisani, A., Oxidative stress and synaptic dysfunction in rodent models of Parkinson's disease. Neurobiol Dis 2022, 173.
69.Mailloux, R. J., An Update on Mitochondrial Reactive Oxygen Species Production. Antioxidants (Basel) 2020, 9, (6).
Point 3: In your work, sufficient attention is paid to the connection of disorders in the respiratory chain complexes (especially COX 1) with Parkinson's disease. But these disorders are the basis of OS formation. Please explain what you meant when you say that OXPHOS is a consequence of violations of mitochondrial proteostasis - perhaps it is necessary to replace OXPHOS with OS in the terminology and the corresponding text? (Lines 15, 37, 68). Or you’ve mean OXPHOS violations, depletions, decrease?
Response 3: The description of the section on the role of OS in PD is unclear. The manuscript wanted to describe the role of OS in PD, so OXPHOS was changed to OS.
Point 4: At Line 188-193 you are writing about electron transfer through Respiratory chain (OXPHOS pathway) which results in ROS synthesis. I am afraid it louds not very clear, because the main function of electron transport is not the ROS synthesis, but more complex processes leading to the changing in Gibbs free energy between COX, which makes ATP synthesis possible, when oxygen is the acceptor of electrons, which are needed for this reactions, and electrons leakage from the respiratory chain is the source of ROS as a by-product of OXPHOS. At the same time, the loss of electrons and the formation of ROS occurs under normal conditions, without damage to mitochondria and the formation of OS. In my opinion, this process can be described in your work more accurately.
Response 4: Thanks for sincerely review and consideration. This section wants to describe the adverse effects of excessive ROS. The mitochondrial respiratory chain utilizes a series of electron transfer reactions to generate cellular ATP through OXPHOS. A consequence of electron transfer is the generation of ROS, which contributes to both homeostatic signaling as well as OS during pathology. Excessive ROS production has detrimental effects and maintaining MRC execution is critical to slowing the pathological process of the disease. We add the description of this viewpoint in Line 207-226.
" Mitochondrial respiratory chain (MRC) impairment has been well-documented in the substantia nigra of PD patients, as well as in various animal models. Considering the detrimental effects of ROS over-production and inadequate neuronal energy supply, maintaining MRC execution is critical to slowing the pathological process of PD [70-72]. The MRC is the basic structure of OXPHOS and plays a central role in cellular energy metabolism. The MRC consists of four enzyme complexes, including nicotinamide adenine dinucleotide ubiquinone reductase (NADH dehydrogenase; CI), ubiquinone succinate oxidoreductase (complex II; CII), ubiquinone cytochrome oxidoreductase (complex III; CIII) and cytochrome c oxidase (complex IV; CIV), as well as two mobile electron carriers, coenzyme Q (CoQ) and cytochrome c (Cytc) [73]. Increased stress due to ROS production is one of the proposed mechanisms of dopaminergic neuronal death in PD, and mitochondrial complex I is thought to be one of the main sources of ROS [36]. It has long been known that patients with idiopathic PD have reduced disease-specific mitochondrial CI activity or protein levels in the substantia nigra after death [74]. Many PD-related genes (e.g., PINK1, Parkin, DJ-1, LRRK2, MNRR1, SNCA, and VPS35) interact with or contribute to the assembly, phosphorylation, or normal activity of CI sub-units [75]. It has been found that CI activity in the substantia nigra and CI sub-unit levels in the striatum were reduced in PD patients. A decrease in CI has also been detected in the muscle and blood cells of PD patients. In addition, reductions in other MRC complexes have been reported in many PD patients [76]."
Reference:
36.Malpartida, A. B.; Williamson, M.; Narendra, D. P.; Wade-Martins, R.; Ryan, B. J., Mitochondrial Dysfunction and Mitophagy in Parkinson's Disease: From Mechanism to Therapy. Trends Biochem Sci 2021, 46, (4), 329-343.
70.Li, Y.; Yang, C.; Wang, S.; Yang, D.; Zhang, Y.; Xu, L.; Ma, L.; Zheng, J.; Petersen, R. B.; Zheng, L.; Chen, H.; Huang, K., Copper and iron ions accelerate the prion-like propagation of α-synuclein: A vicious cycle in Parkinson's disease. International journal of biological macromolecules 2020, 163, 562-573.
71.Goncalves, A. M.; Pereira-Santos, A. R.; Esteves, A. R.; Cardoso, S. M.; Empadinhas, N., The Mitochondrial Ribosome: A World of Opportunities for Mitochondrial Dysfunction Toward Parkinson's Disease. Antioxid Redox Sign 2021, 34, (8), 694-711.
72.Martin-Jimenez, R.; Lurette, O.; Hebert-Chatelain, E., Damage in Mitochondrial DNA Associated with Parkinson's Disease. DNA Cell Biol 2020, 39, (8), 1421-1430.
73.Kalpage, H. A.; Wan, J. M.; Morse, P. T.; Zurek, M. P.; Turner, A. A.; Khobeir, A.; Yazdi, N.; Hakim, L.; Liu, J.; Vaishnav, A.; Sanderson, T. H.; Recanati, M. A.; Grossman, L. I.; Lee, I.; Edwards, B. F. P.; Huttemann, M., Cytochrome c phosphorylation: Control of mitochondrial electron transport chain flux and apoptosis. Int J Biochem Cell B 2020, 121.
74.Dar, G. M.; Ahmad, E.; Ali, A.; Mahajan, B.; Ashraf, G. M.; Saluja, S. S., Genetic aberration analysis of mitochondrial respiratory complex I implications in the development of neurological disorders and their clinical significance. Ageing research reviews 2023, 87, 101906.
75.Cabral-Costa, J. V.; Kowaltowski, A. J., Neurological disorders and mitochondria. Mol Aspects Med 2020, 71, 100826.
76.Jimenez-Salvador, I.; Meade, P.; Iglesias, E.; Bayona-Bafaluy, P.; Ruiz-Pesini, E., Developmental origins of Parkinson disease: Improving the rodent models. Ageing research reviews 2023, 86, 101880.
Point 5: It is known that activation of tissue respiration in the respiratory chain leads to an increase not only in ATP synthesis, but also in ROS production. An increased level of ROS can, on the one hand, strengthen certain mechanisms of RedOx regulation, but, on the other hand, cause the development of OS. Please explain what level of improvement of OXPHOS you mean. An increase in weakened OXPHOS with PD to a normal level? At Line 310 you claim that exercise can inhibit OS. But what is the underlying mechanism of OS inhibition if OXPHOS is activated after exercises and ROS production must be increased, too?
Response 5: The paragraph is not clearly expressed. Exercise can prevent damage from mitochondrial oxidative stress by modulating impaired MRC complexes and controlling the overproduction of ROS. We add the description of this viewpoint in Line 297-312.
" The levels of CIV and MRC complexes were higher in the skeletal muscle of older subjects who had undergone intense exercise training than those who had not, suggesting that exercise training improves mitochondrial function and the mitochondrial network structure in skeletal muscle of ageing humans [97, 98]. Twelve weeks of resistance exercise training resulted in qualitative and quantitative changes in skeletal muscle mitochondrial respiration and moderate changes in mitochondrial proteins; particularly CI activity. The capacity for OXPHOS and electron transport systems is greatly reduced in sedentary animals, and the use of resistance exercise may prevent harmful conditions in mitochondrial skeletal muscle function [99]. Exercise-induced increases in CII activity may eliminate succinate, further reducing ROS production by platelet mitochondria and ultimately inhibiting systemic OS in patients with cardiovascular disease. Furthermore, low-intensity blood flow-limiting resistance exercise increased coupled mitochondrial respiration in skeletal muscle to resist OS by increasing CII activity in patients with heart failure [100]. Exercise training significantly protected endothelial cells from oxidative damage caused by hyperhomocysteinemia and prevented the development of atherosclerosis by activating SIRT1 and inhibiting OS [101, 102]."
Reference:
97.Wyckelsma, V. L.; Levinger, I.; McKenna, M. J.; Formosa, L. E.; Ryan, M. T.; Petersen, A. C.; Anderson, M. J.; Murphy, R. M., Preservation of skeletal muscle mitochondrial content in older adults: relationship between mitochondria, fibre type and high-intensity exercise training. J Physiol-London 2017, 595, (11), 3345-3359.
98.Ringholm, S.; Gudiksen, A.; Frey Halling, J.; Qoqaj, A.; Meizner Rasmussen, P.; Prats, C.; Plomgaard, P.; Pilegaard, H., Impact of Aging and Lifelong Exercise Training on Mitochondrial Function and Network Connectivity in Human Skeletal Muscle. J Gerontol A Biol Sci Med Sci 2023, 78, (3), 373-383.
99.Marin, C. T.; de Souza Lino, A. D.; Avelar, I. D. S.; Barbosa, M. R.; Scarlato, G. C. G.; Cavalini, D. F.; Tamanini, F.; Alexandrino, A. V.; Vercesi, A. E.; Shiguemoto, G. E., Resistance training prevents dynamics and mitochondrial respiratory dysfunction in vastus lateralis muscle of ovariectomized rats. Exp Gerontol 2023, 173, 112081.
100.Groennebaek, T.; Sieljacks, P.; Nielsen, R.; Pryds, K.; Jespersen, N. R.; Wang, J.; Carlsen, C. R.; Schmidt, M. R.; de Paoli, F. V.; Miller, B. F.; Vissing, K.; Botker, H. E., Effect of Blood Flow Restricted Resistance Exercise and Remote Ischemic Conditioning on Functional Capacity and Myocellular Adaptations in Patients With Heart Failure. Circ-Heart Fail 2019, 12, (12).
101.Lim, A. Y.; Chen, Y. C.; Hsu, C. C.; Fu, T. C.; Wang, J. S., The Effects of Exercise Training on Mitochondrial Function in Cardiovascular Diseases: A Systematic Review and Meta-Analysis. Int J Mol Sci 2022, 23, (20).
102.Chan, S. H.; Hung, C. H.; Shih, J. Y.; Chu, P. M.; Cheng, Y. H.; Lin, H. C.; Hsieh, P. L.; Tsai, K. L., Exercise intervention attenuates hyperhomocysteinemia-induced aortic endothelial oxidative injury by regulating SIRT1 through mitigating NADPH oxidase/LOX-1 signaling. Redox Biol 2018, 14, 116-125.
Point 6: Exercises restores OXPHOS capacity and activates COX protein activity (Line 303-304, 324). Where is the measure when exercises are useful, and where do they become harmful (how is overtraining, excessive exercise, fatigue related to the state of mitochondria and proteostasis?) I think that a discussion of these issues would be useful in your article.
Response 6: The upper limit of the amount of exercise associated with beneficial therapeutic effects has not yet been clearly established, so only a brief mention of the importance of moderate exercise is made in this paragraph and in the conclusions section. We add the description of this viewpoint in Line 313-323, 444-451.
" There is an upper limit to the amount of intensive exercise that can be performed without disrupting metabolic homeostasis, beyond which—for example, after a period of progressively harder training that exceeds expected physiological changes (i.e., over-training)—negative effects on metabolic health and physical adaptation begin to emerge, which appear to be caused by a partial shutdown of mitochondrial respiration and H2O2 production [103]. In animal models, moderate exercise increases the synthesis of OXPHOS complexes, improves mitochondrial energy efficiency, and improves brain mitochondrial bioenergetic function through mitochondrial enzyme induction. However, strenuous exercise leads to a significant reduction in OXPHOS CIV, a decrease in mitochondrial OXPHOS, and a decrease in ATP production [104]. Moderate exercise to maintain mitochondrial homeostasis at normal levels is, thus, essential."
" Although there has been a great deal of research on the prevention and treatment of PD through exercise, there has been little systematic research on whether mitochondrial proteostasis plays a key role in delaying and improving PD through exercise, which is expected to facilitate the development of appropriate exercise prescriptions. Our future research should deepen the role and mechanism of mitochondrial proteostasis in delaying and improving PD through exercise in order to provide new ideas and a theoretical basis for the prevention and treatment of PD through exercise, as well as helping those in the medical field to develop relevant and rational exercise prescriptions."
Reference:
103.Flockhart, M.; Nilsson, L. C.; Tais, S.; Ekblom, B.; Apro, W.; Larsen, F. J., Excessive exercise training causes mitochondrial functional impairment and decreases glucose tolerance in healthy volunteers. Cell Metab 2021, 33, (5), 957-970 e6.
104.Sanguesa, G.; Batlle, M.; Munoz-Moreno, E.; Soria, G.; Alcarraz, A.; Rubies, C.; Sitja-Roqueta, L.; Solana, E.; Martinez-Heras, E.; Meza-Ramos, A.; Amaro, S.; Llufriu, S.; Mont, L.; Guasch, E., Intense long-term training impairs brain health compared with moderate exercise: Experimental evidence and mechanisms. Ann N Y Acad Sci 2022, 1518, (1), 282-298.
Reviewer 3 Report
This review by Li et al covers the evidence suggesting that physical exercise might be beneficial to Parkinson's disease (PD) by virtue of its impact on mitochondrial proteostasis. Although a reasonable review of the literature on this topic, unfortunately the English grammar renders the review difficult to interpret and, in places, causes misstatements.
1. The title implies that exercise induces mitochondrial proteostasis. Proteostasis is the defaults condition, however, not something that needs to be "induced." A better title might be, for example: The Role of Exercise in Maintaining Mitochondrial Proteosasis in Parkinson's Disease
2. Some of the fonts in the figures are far too small to be legible.
3. Some protein names seem to have been "lost in translation". For example, in lines 175-176, TIM23 is defined as "translocate of endosomal." There is no such protein. The recommended name is "Mitochondrial import inner membrane translocase subunit TIM23".
4. There are strange usages of language in many places. To cite just one example, the second sentence of Subsection 5.2 starts as "The study...". The previous sentence begins "Studies..." but only cites one reference. Perhaps the authors are meaning to refer to just one study, but it is unclear to the reader as written.
To cite just one more random example, Lines 411-412 in the Discussion section reads "Future research should deepen the role and mechanism of mitochondrial proteostasis...." The sentence should probably read something like "...deepen our understanding of the role...."
5. References are missing, such as for the sentence ending on Line 354.
6. It is not clear why there is an Appendix that is added before the references section. It should either be included as a figure or placed into Supplementary Materials.
7. Most of all, there are grammatical errors throughout the manuscript that make the reading difficult. For example, The second sentence of section 2 (lines 46-48) talks about mitochondria (the plural form of mitochondria) in the singular. It would be fine to talk about "the mitochondrion" but the use of "It" after a sentence describing mitochondria is not appropriate. There are too many other examples than is possible to elaborate upon.
Author Response
Response to Reviewer’s Comment
First of all, we would like to thank the reviewers for their invaluable suggestions and positive comments on the manuscript. We carefully examined these comments and have revised the manuscript point-by-point along the lines suggested.
The corrections in the revised manuscript are marked using the 'track changes' function.
Although a reasonable review of the literature on this topic, unfortunately the English grammar renders the review difficult to interpret and, in places, causes misstatements.
Response: We thank the reviewers for their valuable suggestions and this manuscript has been edited using the English editing service provided by MDPI.
Point 1: The title implies that exercise induces mitochondrial proteostasis. Proteostasis is the defaults condition, however, not something that needs to be "induced." A better title might be, for example: The Role of Exercise in Maintaining Mitochondrial Proteosasis in Parkinson's Disease
Response 1: After taking your suggestion fully into account, we have revised the title to read: The Role of Exercise in Maintaining Mitochondrial Proteostasis in Parkinson's Disease
Point 2: Some of the fonts in the figures are far too small to be legible.
Response 2: We have re-adjusted the figures and have included them at the end of the manuscript.
Point 3: Some protein names seem to have been "lost in translation". For example, in lines 175-176, TIM23 is defined as "translocate of endosomal." There is no such protein. The recommended name is "Mitochondrial import inner membrane translocase subunit TIM23".
Response 3: We have checked the full text and made corrections where errors have occurred.
Point 4: There are strange usages of language in many places. To cite just one example, the second sentence of Subsection 5.2 starts as "The study...". The previous sentence begins "Studies..." but only cites one reference. Perhaps the authors are meaning to refer to just one study, but it is unclear to the reader as written.
To cite just one more random example, Lines 411-412 in the Discussion section reads "Future research should deepen the role and mechanism of mitochondrial proteostasis...." The sentence should probably read something like "...deepen our understanding of the role...."
Response 4: The two issues mentioned by the reviewer have been revised. "The study..." and "Studies..." have been revised. Line 369-372, "Studies have demonstrated the protective effect of exercise on mitochondrial dysfunction in PD [113], and have found that platform running improved symptoms in PD mice and increased mitophagy activity, as evidenced by reduced levels of the mitophagy detection proteins PINK1, Parkin, and p62 [114-116]. "
Point 5: References are missing, such as for the sentence ending on Line 354.
Response 5: As the descriptions of these sentences are taken from a single reference, no reference has been added. We have added the reference. Line 385-394. "Regular exercise can limit the loss of dopaminergic neurons and improve mitochondrial dysfunction, thereby alleviating PD [63]. It has been found that the expression levels of TOM40, TOM20, and TIM23 were significantly reduced in the MPTP group of mice, and that increased expression of α-syn resulted in decreased expression of MIM-related proteins. Inactivation of MIM through α-syn accumulation blocks the import of mitochondrial precursor proteins responsible for mitochondrial structure and function [63]. The expressions of TOM40, TOM20, and TIM23 were significantly increased in mice after exercise, and the reduction of α-syn induced by treadmill exercise led to an increase in the expression of MIM-related proteins, thus increasing the import of mitochondrial precursor proteins and improving mitochondrial function [63]. "
Reference:
Koo, J. H.; Cho, J. Y.; Lee, U. B., Treadmill exercise alleviates motor deficits and improves mitochondrial import machinery in an MPTP-induced mouse model of Parkinson's disease. Exp Gerontol 2017, 89, 20-29.
Point 6: It is not clear why there is an Appendix that is added before the references section. It should either be included as a figure or placed into Supplementary Materials.
Response 6: The images inserted in this section are in accordance with the regulations outlined in the manuscript submission overview. The journal requires the following: Supplementary Materials: Describe any supplementary material published online alongside the manuscript (figure, tables, video, spreadsheets, etc.). Please indicate the name and title of each element as follows Figure S1: title, Table S1: title, etc.
Point 7: Most of all, there are grammatical errors throughout the manuscript that make the reading difficult. For example, The second sentence of section 2 (lines 46-48) talks about mitochondria (the plural form of mitochondria) in the singular. It would be fine to talk about "the mitochondrion" but the use of "It" after a sentence describing mitochondria is not appropriate. There are too many other examples than is possible to elaborate upon.
Response 7: The manuscript has been edited using the English editing service provided by MDPI.

Round 2
Reviewer 3 Report
The authors made a good effort to improve the readability of the manuscript. I recommend publishing in its current form.